# Systematic Review of the Relationships between 24-Hour Movement Behaviours and Health Indicators in School-Aged Children from Arab-Speaking Countries

**DOI:** 10.3390/ijerph18168640

**Published:** 2021-08-16

**Authors:** Yazeed A. Alanazi, Eduarda Sousa-Sá, Kar Hau Chong, Anne-Maree Parrish, Anthony D. Okely

**Affiliations:** 1Early Start and School of Health & Society, University of Wollongong, Wollongong, NSW 2522, Australia; emdsr885@uowmail.edu.au (E.S.-S.); khc745@uowmail.edu.au (K.H.C.); aparrish@uow.edu.au (A.-M.P.); tokely@uow.edu.au (A.D.O.); 2Research Centre in Physical Activity, Health and Leisure, Faculty of Sport, University of Porto, 4200-450 Porto, Portugal; 3CIDEFES—Centro de Investigação em Desporto, Educação Física e Exercício e Saúde, Universidade Lusófona, 1749-024 Lisbon, Portugal; 4Laboratory for Integrative and Translational Research in Population Health, 4050-091 Porto, Portugal; 5Illawarra Health and Medical Research Institute, University of Wollongong, Wollongong, NSW 2522, Australia

**Keywords:** movement behaviours, child, sleep, sedentary behaviour, physical activity, Arab

## Abstract

The Australian and Canadian 24-hour movement guidelines for children and youth synthesized studies in English and French or other languages (if able to be translated with Google translate) and found very few studies published in English from Arabic countries that examined the relationship between objectively measured sedentary behaviour (SB), sleep and physical activity (PA) and health indicators in children aged 5–12 years. The purpose of this systematic review was to investigate the relationships between 24-hour movement behaviours and health indicators in school-aged children from Arab-speaking countries. Online databases MEDLINE, EMBASE, SPORTdiscus, CINAHL, PsycINFO and Scopus were searched for English, French and Arabic studies (written in English), while Saudi Digital Library, ArabBase, HumanIndex, KSUP, Pan-Arab Academic Journal, e-Marefa, Al Manhal eLibrary and Google Scholar were searched for Arabic studies. The Grading of Recommendations Assessment, Development and Evaluation framework was used to assess the risk of bias and the quality of evidence for each health indicator. A total of 16 studies, comprising 15,346 participants from nine countries were included. These studies were conducted between 2000 and 2019. In general, low levels of PA and sleep and high SB were unfavourably associated with adiposity outcomes, behavioural problems, depression and low self-esteem. Favourable associations were reported between sleep duration and adiposity outcomes. SB was favourably associated with adiposity outcomes, withdrawn behaviour, attention and externalizing problems. PA was favourably associated with improved self-esteem and adiposity outcomes. Further studies to address the inequality in the literature in the Arab-speaking countries to understand the role of 24-hour movement behaviours and its positive influence on health outcomes across childhood are urgently needed.

## 1. Introduction

For years, movement guidelines for children and adolescents have concentrated on moderate- to vigorous-intensity physical activity (MVPA) [1]. However, focusing on PA per se and omitting other movement behaviours, such as SB and sleep, has reduced our perception of how these daily movement behaviours interact to affect children’s health [2]. Hence, an approach that integrates all components of movement behaviours is required to review their combined influence on health and development [2].

In 2016, the Canadian Society for Exercise Physiology released the world’s first integrated 24-hour movement guidelines for children and youth (5–17 years) [3]—a new concept describing the integration of PA, SB and sleep over the 24-hour period. This terminology is regarded as a shift in daily movement behaviour research and illustrates an evolution in PA guidelines [3]. These guidelines were launched based on the results of four systematic reviews, investigating associations between PA, SB, sleep, and movement combinations, each one with health indicators [3]. The authors found that children’s total PA was favourably associated with physical, psychological/social, and cognitive health indicators [4]. Higher levels of TV and screen time viewing were associated with unfavourable body composition, cardio-metabolic disease risk scores, hostile behavioural conduct/pro-social behaviour indicators, lower fitness and self-esteem in children [5]. Shorter sleep duration was associated with poorer health outcomes [6]. Children who had higher levels of PA and sleep and less SB had more desirable measures of adiposity and cardiometabolic health when compared with those with a combination of low PA and low sleep and high SB. Similarly, those with high PA and high sleep or high PA and low SB profiles demonstrated favourable health indicators compared with low PA and low sleep, or low PA and high SB profiles [7].

In 2018, the literature on 24-hour movement guidelines was updated as part of the development of Australian 24-Hour Movement Guidelines for Children and Young People [8]. These guidelines were launched as evidence-based guidelines to address movement behaviours observed over the whole day instead of focusing on these behaviours in isolation. Previous studies only captured specific movement behaviours (e.g., MVPA) during waking hours—which accounts for a small portion of children’s daily activity (<5%) in the 24-hour period; while sleep (~40%), SB (~40%) and LPA (~15%) make up nearly 95% of the day [2]. Furthermore, these guidelines were launched based on the results of systematic reviews that synthesized studies in English and French or other languages if able to be translated with Google translate, and found very few studies published in English or French from Arabic countries that examined the relationship between objectively measured SB, sleep and PA and health indicators in children aged 5–12 years. The search criteria did not elicit any studies published in Arabic from Arab countries.

Even though more than 140 million children live in the 22 Arabic countries [9], where there are high and increasing levels of childhood obesity and sedentary behaviour [10,11], the lack of evidence from these countries confirms that research is needed to address this gap in the literature, to understand the role of 24-hour movement behaviours and its influence on important health outcomes (obesity, executive functions, motor development and bone health) across the years of primary school. Moreover, it is likely there may be cultural differences in 24-hour movement behaviours in Arabic countries compared to Western countries, necessitating a separate review. Therefore, the purpose of this systematic review was to investigate the relationship between 24-hour movement behaviours and health indicators in school-aged children in Arabic countries.

## 2. Methods

This systematic review was registered with the International Prospective Register of Systematic Reviews (PROSPERO; Registration no. CRD42020143101). It was conducted and reported following the Preferred Reporting Items for Systematic Reviews and Meta-Analyses (PRISMA) statement for reporting systematic reviews and meta-analyses [12]. The protocol of this study was adopted partly from the systematic review performed by Saunders et al. [7].

### 2.1. Eligibility Criteria

Eligible participants included apparently healthy children aged 5 to 12 years old. Overweight and/or obese children were also included. Studies where the sample were aged above 12 years or below 5 years were included if the mean age was between 5–12 years. To be included, studies had to be peer-reviewed, published, written in Arabic, English or French and reported subjective or objective measurement of PA or SB or sleep or their combination. Grey literature, student dissertations or conference abstracts were excluded. The main outcomes were adiposity, cardiometabolic biomarkers, fitness, behavioural conduct/pro-social behaviour, emotional regulation/psychological distress, cognition (academic achievement), quality of life and injuries. Secondary outcomes included bone density, motor skill development and self-esteem. The review was limited to full manuscripts. There was no minimum sample size. All study designs were included.

Twenty-four-hour movement behaviours incorporate sleep, SB and PA, which are independently defined as:

Sleep: “a naturally recurring state of body and mind characterized by altered consciousness, relatively inhibited sensory activity, inhibition of nearly all voluntary muscles and reduced interactions with surroundings”.[13]

SB: “any waking behaviour characterized by an energy expenditure ≤1.5 metabolic equivalents (METs), while in a sitting, reclining or lying posture”.[14]

PA: “any bodily movement produced by skeletal muscles that results in energy expenditure above the resting metabolic rate”.[15]

### 2.2. Search Strategy

Six electronic databases were searched from January, 1990 to January, 2021 to identify relevant articles that were written in English or French: MEDLINE, EMBASE, SPORTdiscus, CINAHL, PsycINFO and Scopus. Eight electronic databases were searched for Arabic studies: Saudi Digital Library, ArabBase, Human Index, KSUP, Pan-Arab Academic Journal, e-Marefa, Al Manhal eLibrary and Google Scholar. Search terms can be seen in the Appendix A.

### 2.3. Data Extraction

Studies were imported into Endnote X9 software (Thomson Reuters, San Francisco, CA, USA). After de-duplication, three authors (YA, ESS and KHC) screened titles and abstracts for relevant studies. Full-text copies of the eligible studies were assessed for final inclusion. Any disagreement between the three authors was resolved through a discussion and, when necessary, included a fourth author. The reference lists of all included studies were screened for additional studies not listed in the database search. Data were extracted for each study using an Excel spread sheet; each study included article, author, study design, publication year, location, sample size, age, mean age, gender, outcomes and measures, study instrument and results (Table 1).

### 2.4. Quality Assessment

Three authors (YA, ESS and KHC) independently assessed the risk of bias (ROB) using the GRADE framework (Grading of Recommendations Assessment, Development, and Evaluation), which was also used to assess the quality of evidence for each health indicator. GRADE does not have an official tool for assessing ROB in observational studies but recommends the types of study characteristics to be evaluated [16]. The quality of evidence was assessed for each included study design based on selection bias, attrition bias, detection bias, performance bias, and selective reporting bias. Quality of evidence scores were considered “low” for experimental and observational studies. Scores above 6/8 were considered as having low risk of bias.

**Table 1 ijerph-18-08640-t001:** Characteristics of included studies.

Literature Reference and Country	Study Design	Sample Size (% Female), Mean Age or Age Range (Years)	Type of Behaviour	Exposure and Assessment Instrument	Outcomes	Statistical Analysis & Confounders (If Reported)	Main Results
AlHazzaa et al., (2019) Saudi Arabia [17]	Cross-sectional	1033 (51.1% female); mean age = 9.2 ± 1.7	Sleep	Sleep: Parent-proxy reported average sleep duration per night (<9 h vs. ≥9 h).	Adiposity: body weight (kg) and BMI.	Logistic regression analysis. Confounders: body weight, age and gender.	No significant association between sleep duration and overweight or obesity status (aOR = 1.00; 95% CI 0.71 t0 1.64; *p* = 0.717).
Al-Hazzaa, (2007) Saudi Arabia [18]	Cross-sectional	296 (100% male); mean age = 10.3 ± 1.3	PA	PA: Pedometer measured steps taken/day.	Adiposity: BMI, skinfold measurements (triceps and subscapular, body fat %, FMI and FFMI.	Pearson’s correlation. Confounders: age, gender, daily pedometer counts and total energy expenditure.	Significant negative associations between step counts/day and body fat % (r = −0.207; *p* = 0.006), BMI (r = −0.198; *p* = 0.007), FMI (r = −0.214; *p* = 0.004), but not with FFMI (r = −0.089; *p* = 0.231).
Hassan and Al-Kharusy, (2000) Oman [19]	Pilot study	109 (100% male); mean age = 9.68 ± 0.92	PA and SB	PA: Leisure time sport activities personal activity score (hours/week) assessed with 1.6–km run/walk. SB: Parent-proxy reported duration of TV watching and/or playing video or computer games.	Fitness: cardiorespiratory endurance. Adiposity: Log sum of 5 skinfold measurements (triceps, subscapular, suprailiac, abdominal and thigh).	Pearson correlation coefficients.	Personal activity score has a strong negative correlation with the time to complete the 1.6 km run/walk and the sum of skinfolds (*r* = −0.40, −0.42; *p* = 0.001). No significant associations between TV watching hours and fitness or fatness (*p* = *n*.r).
Hadhood et al., (2016) Egypt [20]	Cross-sectional	711 (54.5% female); mean age = 10.36 ± 1.9	PA	PA: Parent-proxy reported weekly practice of physical exercise.	Adiposity: BMI and body weight.	Chi square test.	No significant association between physical exercise and overweight and/or obesity (*p* = 0.19).
Badawi et al., (2013) Egypt [21]	Cross-sectional	852 (50.2% female); mean age = 9.5 ± 1.8	PA and SB	PA: Parent-proxy reported practice of sports, and transportation to school. SB: Parent-proxy reported time spent watching TV.	Adiposity: BMI and body weight.	t-test, ANOVA test.	Significant association between low PA and BMI (*p* = < 0.001). Significant association between SB and BMI (*p* = < 0.001).
Al-Lahham et al., (2019) Palestine [22]	Cross-sectional	1320 (48% female); mean age = 9.5 ± 1.5	PA and SB	PA: Parent-proxy reported daily PA (min), mode of transport to school. SB: Parent-proxy reported screen time (min).	Adiposity: BMI and body weight.	Chi square test. Confounders: transporting means to school, total screen time, total PA time and age.	Significant association between levels of PA (transportation means only) and BMI (*p* = 0.031). Screen time had no significant effect on BMI, however, it had a borderline effect (*p* = 0.069).
Jemaa et al., (2018) Tunisia [23]	Cross-sectional	40 (47.5% female); mean age = 9.34 ± 0.94	PA and SB	PA and SB: Accelerometer measures (LPA, MPA, VPA, MVPA); Subjective measures (mean PA Questionnaire for Older Children (PAQ-C) score and intensity classification).	Adiposity: % fat mass.	Pearson Correlation coefficient.	The average MVPA showed a negative significant correlation with body fat % (r = −0.343, *p* = 0.030). The score of PA determined by PAQ-C was not significantly correlated with the body fat % (r = −0.227, *p* = 0.158).
Lafta and Kadhim, (2005) Iraq [24]	Case control	2084 (male and female); 7–13 (age range)	SB	SB: Parent proxy reported watching TV (>3 h/day) via questionnaire.	Adiposity: BMI-defined overweight/obese.	Chi-square test. Confounders: age, birth rank, type of feeding during infancy, dietary pattern, pattern of PA and working after school time.	Watching TV (> 3 h/day) was a significant factor for overweight in 7–9 year males ($\chi^{2}$ = 19.69, 95% CI 1.79 to 4.97; *p* < 0.001).
Alam (2008) Saudi Arabia [25]	Cross-sectional	1072 (100% female); 8–12 (age range)	SB	SB: Parent proxy reported duration of TV watching via questionnaire.	Adiposity: BMI and body weight.	Chi square test.	Watching TV (>2 h/day) was significantly higher among obese students ($\chi^{2}$ = 12.98, *p* = 0.011).
Arora et al., (2018) Qatar [26]	Cross-sectional	264 (62.1% female); mean age = 9.0 ± 1.2	Sleep and SB	Sleep: weekday sleep duration. SB: SB time assessed with wrist Actigraphy/Technology Use Questionnaire.	Adiposity: BMI z-score, waist circumference, neck circumference, body fat % and fat mass.	Multiple linear regression. Confounders: objective estimate of sedentariness, dietary habits, age, sex, ethnicity and total technology use.	Significant associations between sleep duration and sleep insufficiency (<8 h) and all indicators of obesity (*p* < 0.001) except for neck circumference. Waist circumference (cm) yielded the largest effect: β = −4.99, *p* < 0.001 (average sleep duration) and β = 6.49, *p* < 0.001 (<8 h). Sleep duration variation (night-to-night sleep duration variability) was not significantly associated with any outcome. Poor sleep efficiency was positively associated with body fat percentage (β = 2.20, *p* = 0.028).
Al-Kutbe et al., (2017) Saudi Arabia [27]	Cross-sectional	266 (100% female); 8–11 (age range)	PA and SB	PA and SB: Number of steps taken/day with accelerometer (WGT3X-BTActigraph).	Adiposity: body weight (kg).	Multiple linear regression. Confounders: daily energy intake, daily total energy expenditure, body weight, age and family income.	No association between the number of steps or the time spent in MVPA and body weight (Beta = 0.034; *p* = 0.575, 0.368).
Al-Hazzaa and Alrasheedi, (2007) Saudi Arabia [28]	Cross-sectional	224 (51.3% female); mean age = 5.19 ± 0.85	PA and SB	PA: Pedometer measured steps taken/day. SB: Parent proxy reported duration of TV watching/day via questionnaire.	Adiposity: BMI, skinfold measurements (triceps, subscapular (sum and ratio), FM %, FFM %, FMI and FFMI.	One-way ANOVA and post hoc test (Scheffe). Confounders: body size for FMI and FFMI only.	No significant differences between obese and non-obese children in steps counts/day (*p* = 0.109). No significant difference between active and inactive preschool children in any of the measured anthropometric and body composition variables (body weight (*p* = 0.644), BMI (*p* = 0.961), triceps skinfold (*p* = 0.975), subscapular skinfold (*p* = 0.738), sum of 2 skinfolds (*p* = 0.854), subscapular/triceps ratio (*p* = 0.219), fat % (*p* = 0.985), fat mass (*p* = 0.664), fat free mass (*p* = 0.744), FMI (*p* = 0.850), FFMI (*p* = 0.896). Obese children spent significantly more time watching TV (197.5 ± 89.3 min/day) than their non-obese peers (150.0 ± 60.9 min/day) (*p* = 0.001).
Alqaderi et al., (2016) Kuwait [29]	Longitudinal study	8317 in 1st phase and 6316 in 2nd phase (61.4% female); 8–11 at visit 1, 10–12 at follow up (age range)	Sleep	Sleep: Lifestyle habits interview reported daily sleep hours, TV and video game use.	Adiposity: Waist circumference.	Multilevel longitudinal linear regression model. Confounders: age and gender.	Short sleep duration was significantly associated with increased waist circumference (beta = −0.11; 95%CI 0.14 to 0.17; *p* = < 0.05).
Al-Ghamdi, (2013) Saudi Arabia [30]	Case control	397 (49.3% female); mean age = 11.4 (SD: *n*.r.)	PA and SB	PA and SB: questionnaire (interview) reported watching TV (>3 h/day) and daily exercise.	Adiposity: BMI.	Chi-square test. Confounders: TV, VG time/day, age, daily exercise/day.	Watching TV (>3 h/day), especially over the weekend, was significantly associated with childhood obesity ($\chi^{2}$ = 4.136, *p* = 0.042). No significant associations between the rate of exercising at school, home and outdoors and obesity ($\chi^{2}$ = 1.248, 1.032, 2.604; *p* = 0.870, 0.905, 0.626).
Yousef et al., (2013) UAE [31]	Cross-sectional	197 (34% female); mean age = 8.7 ± 2.1	SB	SB: Parent proxy reported watching TV (>2 h/day).	Behavioral problems.	Chi square test, logistic regression. Confounders: birth order and number of siblings.	Watching TV/video game > 2 h/day was associated with withdrawn behavior (OR = 0.275; 95% CI 0.106 to 0.712; *p* = 0.008), attention problem (OR = 0.480; 95% CI 0.241 to 0.956; *p* = 0.037), externalizing problems (OR = 0.393; 95% CI 0.201 to 0.771; *p* = 0.007) and Child Behavior Checklist total score (OR = 0.441; 95% CI 0.229 to 0.848; *p* = 0.014).
Zayed and Kilani, (2014) Oman [32]	Cross-sectional	165 (100% female); 10–13 (age range)	PA	PA: number of occurrences and the duration of the practice of PA per week assessed with PA interview questionnaire.	Depression and low self-esteem.	One-way ANOVA and post hoc test (Scheffe).	Regular PA was significantly associated with improved self-esteem; differences were seen between those who never exercised and those who exercised regularly (mean square = 358.257; F = 4.787; *p* = 0.10).

Abbreviations: SD = standard deviation; BMI = body mass index; FMI = fat mass index; FM = fat mass; FFM = fat-free mass; WC = waist circumference; PA = physical activity; MVPA = moderate-to-vigorous physical activity; SB = sedentary behavior; n.r. = not reported; OR = odds ratio; CI = confidence interval.

## 3. Results

A total of 612 studies were eligible for inclusion. After title and abstract screening, 102 studies were assessed for full-text review. Of those, 86 were excluded for the following reasons: did not contain measures of PA or SB or sleep as an independent variable (*n* = 21); did not contain a measure of a health indicator and its association with PA or SB or sleep (*n* = 13); out of range for age (*n* = 37); dissertation (*n* = 7); studies conducted in non-Arab countries (*n* = 4); unavailability of the full article (*n* = 3); and lack of statistical data (*n* = 1). After all exclusions, 16 studies met the inclusion criteria (Figure 1). These studies provided results from 15,346 participants from 9 Arabic countries: Saudi Arabia, *n* = 6, United Arab Emirates, *n* = 1, Egypt, *n* = 2, Oman, *n* = 2, Kuwait, *n* = 1, Iraq, *n* = 1, Tunisia, *n* = 1, Qatar, *n* = 1 and Palestine, *n* = 1 (Table 1). Of all included studies, 12 were cross-sectional, two were case-control, one was longitudinal and one was a pilot study. These studies were conducted between 2000 and 2019 and included children between 3.4 and 14 years of age (mean age 5.19–11.4 years). Sample sizes ranged from 40 to 8317 participants. Out of the 16 included studies, 14 reported data on adiposity [17,18,19,20,21,22,23,24,25,26,27,28,29,30], one on behavioural problems [31], one on depression and low self-esteem [32] and one on fitness [19]. Out of the sixteen studies included in this review, eight studies (50%) were classified as having a low ROB and eight as having a high ROB (50%). All studies had a reliable and/or valid tool to assess movement behaviours and health outcomes. The criteria used to assess ROB can be seen in Table 2. It was not possible to conduct meta-analyses due to heterogeneity of the data, therefore, narrative syntheses were conducted.

### 3.1. Measurement of Movement Behaviours

Sleep was objectively measured by wrist actigraphy in one study [26] and subjectively measured using questionnaires in two studies [17,29]. SB was measured using screen time in eight studies [19,21,22,24,25,28,30,31] and objectively measured using accelerometers in two studies [23,27], and by both methods (wrist actigraphy and a questionnaire) in one study [26].

PA was measured using an accelerometer in one study [27] and by pedometer in two studies [18,28]; whereas five studies measured it subjectively using parent proxy-reports [20,21,22]. Two studies combined both report and device-based methods: accelerometers plus questionnaire [23] and cardiorespiratory endurance (1.6 km run/walk) plus questionnaire [19].

### 3.2. Health Indicators

#### 3.2.1. Adiposity

As shown in Table 3, adiposity was assessed through the following indicators: BMI, body weight, % fat mass, BMI z-score, and waist circumference. It was reported in 14 studies, of which 10 were cross-sectional [17,18,20,21,22,23,25,26,27,28], one was longitudinal [29], two were case control [24,30] and one was a pilot study [19]. Three studies investigated the relationship between sleep and adiposity outcomes. Of the three studies, two reported significant positive associations [26,29] while one found no significant relationship [17].

Nine studies examined the relationship between SB and adiposity outcomes. SB was positively associated with adiposity outcomes in six of the nine studies [21,24,25,26,28,30]. The remaining three studies found no associations with adiposity outcomes [22,23,27]. Nine studies examined the adiposity relationship with PA. Of the nine studies, five found favourable associations between adiposity outcomes and PA [18,19,21,22,23] while four studies reported null associations [20,27,28,30].

#### 3.2.2. Behavioural Problems

Behavioural problems were reported in only one cross-sectional study [31] involving 197 subjects (mean age 8.7 ± 2.1), which studied the relationship between SB and behavioural problems in school-aged children. The results showed that watching TV/playing VG for more than two hours were positively associated with withdrawn behaviour, attention and externalizing problems.

#### 3.2.3. Depression and Low Self-Esteem

One study examined the association between PA and depression and low self-esteem [32]; it involved 165 female subjects with age range of 10–13 years. The results indicated that regular PA (number of occurrences and the duration of the practice of PA per week) was significantly associated with improved self-esteem.

#### 3.2.4. Fitness

One study assessed the relationship between SB and fitness and reported null associations [19]. The results showed that the personal activity score had a strong negative correlation with the time to complete the 1.6 km run/walk and the sum of skinfolds. There were no significant associations between TV watching hours and fitness or fatness.

There were no studies investigating the associations with the rest of the primary outcomes, namely cardio metabolic biomarkers, psychological distress, cognition (academic achievement), quality of life, injuries, nor on secondary outcomes including bone density and motor skill development.

## 4. Discussion

This study systematically reviewed the relationships between the movement behaviours of physical activity, sedentary behaviours and sleep and health indicators among school-aged children in Arab-speaking countries. Most of the included studies in this review were cross-sectional (75%). The sample sizes ranged from 40 to 8317 participants. These studies reported mostly favourable and some null associations between PA, SB, and sleep and adiposity, behavioural problems, depression and low self-esteem and fitness outcomes. Low levels of PA and sleep and high SB were associated with higher levels of adiposity, behavioural problems, depression and low self-esteem.

Reasons for the small number of studies investigating these movement behaviours in school-aged children in Arab-speaking countries are that this field of research is still in its early stages of development in these countries, with most child health research focusing on more pressing issues such as infectious diseases. In addition, the unstable political environment in some of the Arab countries, has made conducting such research challenging [33]. The availability of funds is another reason that limited the number of these types of studies. For instance, in 2013, the gross domestic expenditure on research and development (GERD) in North America was $427 billion (28.9%) of the worldwide GERD ($1477.7 billion), while Arab countries collectively only spent $15.4 billion (1%) [34]. Finally, although the Arabic databases were searched for relevant studies in this systematic review, there might be some studies that could not be identified due to the small number of Arabic databases available.

Existing Arabic studies assessed movement behaviours in isolation from each other. Of these studies, most used subjective methods to assess PA, sleep and SB. On the other hand, the Canadian [3,4,6,7] and the Australian 24-Hour Movement Guidelines [8] indicated that focusing on movement behaviours across the entire day is more important than focusing on movement behaviours in isolation. For example, a Canadian study investigating the health outcomes associated with meeting the 24-hour movement behaviour guidelines for children and youth showed that meeting none, one and two recommendations were associated with higher BMI z-score, waist circumference, behavioural strengths and difficulties scores and lower aerobic fitness in a gradient pattern (*p*_trend_ < 0.05), while meeting all the guidelines during a 24-hour period was associated with better health [35].

Furthermore, due to the prior emphasis on MVPA [1] and the common use of subjective assessments of PA [4], no study in the present review examined different PA intensities such as light-intensity physical activity (LPA), although emerging evidence suggests that LPA may provide some important health benefits for children and adolescents [2,3,4]. Moreover, children cannot participate in MVPA during all waking hours. Therefore, engaging in LPA (e.g., walking) is considered achievable and an easier way to reduce SB, that also provides health benefits [2].

Few studies in this systematic review assessed the relationship between sleep and adiposity outcomes. A possible explanation of this gap is that sleep and SB are new areas of research in this region when compared to the PA field as the SB included studies were published between 2005–2019, while sleep studies were published in the last five years. Therefore, the Arab countries are urgently in need to conduct more studies that focus on sleep and SB to better understand their impact on school-aged children’s health.

Despite the importance of the weather and its impact on movement behaviours [36], no study in the present review assessed the relationship between 24-hour movement behaviours and climatic factors, although most of the Arab countries have a hot and dry climate [37]. Previous research in other countries indicates that children’s PA levels are affected by seasonal periods across the year and this varies between countries, with PA levels decreasing in specific climatic conditions such as winter, summer, sandstorms areas, humidity and rain [38,39]. Moreover, extreme weather conditions (high or low temperatures) increase SB [40] and decrease sleep efficiency [41], therefore, conducting more studies investigating the association between the weather and movement behaviours and potential interventions may help children in these regions to meet movement behaviour recommendations.

### 4.1. Areas for Future Research

The lack of evidence from the Arab countries confirms that research is needed to address the inequality in the literature, especially with the high and still increasing levels of childhood obesity and SB in the 22 countries. Moreover, it is important to use different types of study designs (longitudinal and experimental) with larger sample sizes to better understand the role of 24-hour movement behaviours, to improve health outcomes. Studies included in this review focused on obesity, however, the field of PA is broader than this health outcome, therefore, it is recommended to conduct more studies reflecting on all movement behaviours across the 24-hour period.

### 4.2. Strengths and Limitations

To our knowledge, the present review is the first study investigating current research assessing the association between movement behaviours and health indicators in school-aged children in the Arab-speaking countries. A lack of Arabic databases is also a potential limitation. Meta-analyses were not possible to conduct due to heterogeneity of the data, therefore, narrative syntheses were conducted.

## 5. Conclusions

Most of the included studies reported favourable associations between movement behaviours and health outcomes. Low levels of PA and sleep and high SB were unfavourably associated with adiposity outcomes, behavioural problems, depression and low self-esteem. Further studies to address the inequality in the literature in the Arab-speaking countries to understand the role of 24-hour movement behaviours and its positive influence on health outcomes across the early years of primary school are urgently needed. Based on the differences between societies and their needs, as well as environmental differences, it might be beneficial to also understand associations between weather conditions and children’s movement behaviours. Conducting more studies on different types/intensities of PA, SB and sleep for both boys and girls, and using different types of study designs (longitudinal and experimental) with larger sample sizes will improve the quality of future studies.

## Figures and Tables

**Figure 1 ijerph-18-08640-f001:**
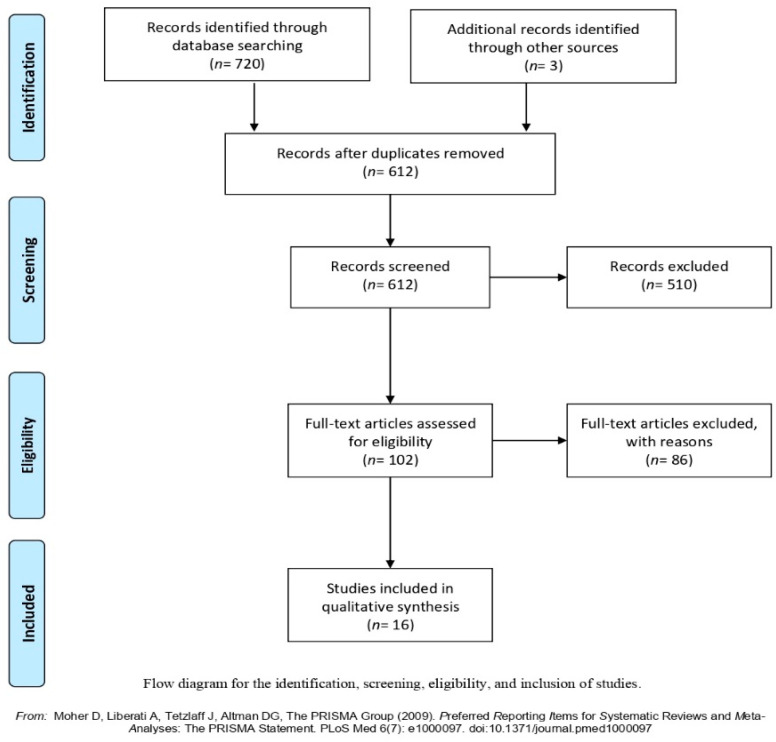
PRISMA Flow Diagram.

**Table 2 ijerph-18-08640-t002:** Risk of bias for included studies.

Study	1. Selection Bias		2. Attrition Bias	3. Detection Bias		4. Performance Bias		5. Selective Reporting Bias	Score *	ROB Quality
	Clear Criteria for Those included and/or Excluded?	Was the Sample Randomly Selected?	Did an Adequate Proportion (At Least 70%) of Those Consenting to Participate in the Study Have Complete Data? (Incomplete Follow-Up; High Loss to Follow-Up; Missing Data)	Did the Study Report the Sources and Details of the Tool Used in the Study to Assess The Outcomes?	Was the Tool Used in the Study to Assess the Outcomes Reliable and/or Valid?	Did the Study Report the Sources and Details of the Measurement Tool Used in the Study for Movement Behaviours?	Were the Measurements of Movement Behaviours in This Study Reliable and/or Valid?	Did the Study Have Complete Data and/or Reports All Outcomes and Not Others Based on the Results?		
	Rating	Rating	Rating	Rating	Rating	Rating	Rating	Rating		
Al-Hazzaa et al., 2019 [17]	1	1	1	1	0	1	0	1	6/8	Low
Al-Hazzaa, 2007 [18]	0	0	0	1	1	1	1	1	5/8	High
Hassan and Al-Kharusy, 2000 [19]	0	0	1	1	1	0	0	1	4/8	High
Hadhood et al., 2016 [20]	1	1	1	1	1	0	0	1	6/8	Low
Badawi et al., 2013 [21]	1	1	1	1	1	0	0	1	6/8	Low
Al-Lahham et al., 2019 [22]	1	0	0	1	1	0	0	1	4/8	High
Jemaa et al., 2018 [23]	1	0	1	1	1	1	1	0	6/8	Low
Lafta and Kadhim, 2005 [24]	1	1	1	1	0	0	0	1	5/8	High
Alam, 2008 [25]	1	0	1	0	1	0	0	1	4/8	High
Arora et al., 2018 [26]	1	0	1	1	1	1	1	1	7/8	Low
Al-Kutbe et al., 2017 [27]	1	0	0	1	1	1	1	1	6/8	Low
Al-Hazzaa and Alrasheedi, 2007 [28]	0	1	1	1	1	1	1	1	7/8	Low
Al-Qaderi et al., 2016 [29]	0	1	1	1	0	1	0	1	5/8	High
Al-Ghamdi, 2013 [30]	1	0	0	1	1	1	0	1	5/8	High
Yousef et al., 2013 [31]	1	1	1	1	1	1	1	1	8/8	Low
Zayed and Kilani, 2014 [32]	0	1	0	1	1	1	0	1	5/8	High
	11/16	8/16	11/16	15/16	13/16	10/16	6/16	15/16		

0 = No or unclear; 1 = Yes. * Scores above 6/8 were considered as having low risk of bias.

**Table 3 ijerph-18-08640-t003:** Results of studies.

Study	Outcomes	Exposure: Favorable Associations			Exposure: Null Associations			Summary
	Adiposity	PA	SB	Sleep	PA	SB	Sleep	
Al-Kutbe et al., 2017	Body weight				✓	✓		5/14 studies showed favorable associations between PA and adiposity outcomes. 4/14 studies showed null associations between PA and adiposity outcomes. 6/14 studies showed favorable associations between SB and adiposity outcomes. 3/14 studies showed null associations between SB and adiposity outcomes. 2/14 studies showed favorable associations between sleep and adiposity outcomes 1/14 studies showed null association between sleep and adiposity outcomes.
AlHazzaa et al., 2019	Body weight and BMI						✓	
Jemaa et al., 2018	% fat mass	✓				✓		
Hadhoodet al., 2016	BMI and body weight				✓			
Badawi et al., 2013	BMI and body weight	✓	✓					
Al-Lahham et al., 2019	BMI and body weight	✓				✓		
Al-Hazzaa, 2007	BMI, skinfold measurements (triceps and subscapular, body fat %, FMI and FFMI.)	✓						
Arora et al., 2018	BMI z-score, waist circumference, neck circumference, body fat % and fat mass.		✓	✓				
Alam, 2008	BMI and body weight		✓					
Al-Hazzaa and Alrasheedi, 2007	BMI, skinfold measurements (triceps, subscapular (sum and ratio), FM %, FFM %, FMI and FFMI.)		✓		✓			
Lafta and Kadhim, 2005	BMI -defined overweight/obese.		✓					
Al-Ghamdi, 2013	BMI		✓		✓			
Alqaderi et al., 2016	Waist circumference			✓				
Hassan and Al-Kharusy, 2000	Log sum of 5 skinfold measurements (triceps, subscapular, suprailiac, abdominal and thigh).	✓						
Yousef et al., 2013	**Behavioural problems**		✓					1/1 studies showed favorable association between SB and behavioural problems.
Zayed and Kilani, 2014	**Depression and low self-esteem**	✓						1/1 studies showed favorable association between PA and depression and low self-esteem.
Hassan and Al-Kharusy, 2000	**Fitness**: cardiorespiratory endurance					✓		1/1 studies showed null association between SB and fitness measures.

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
