# Peer review of "Systematic Review of the Relationships between 24-Hour Movement Behaviours and Health Indicators in School-Aged Children from Arab-Speaking Countries"

_ijerph, 2021, doi:10.3390/ijerph18168640_

Round 1
Reviewer 1 Report
Abstract
- Line 23: why French studies were searched since you focus on Arab-speaking countries?
Introduction:
- Line 51: please add the reference of the sentence “These guidelines were launched based on…”
Methods:
- Search strategy in supplementary information:
“PA” OR “MVPA” OR “LPA” OR “SB” may be included in the search terms
An error in line 295 (media” Sleep*)
Did you used key word search or others?
Results:
- Line 139: 85 or 86 were excluded for the following reasons since you mentioned 85 in the method part but 86 in the figure.
- Double check and modification format and writing errors in Table 1.
Discussion:
- Previous outcomes (adiposity, cardio-metabolic biomarkers, fitness, behavioral conduct/pro-social behavior, emotional regulation/psychological distress, cognition (academic achievement), quality of life and injuries) were not fully discussed compared with the previous articles which focus on other countries.
Conclusion:
- Conclusions in the full text (line 282-283) should include different outcomes instead of using one sentence.
Reviewer 2 Report
Thank you for the opportunity to read this manuscript.
After reading, I have some concerns that I present below.
Page 1, line 43. Delete the word "movement". Physical activity, sedentary behaviours and sleep are three different constructors. Thus, they should not be considered "movement behaviours".
Page 2, lines 83-84. The systematic reviews used were from studies carried out in several countries. Do the authors expect to find different results in school-aged children in Arabic countries?
If so, why?
If not, what is the need to carry out this study?
Page 3, lines 95-97. The search criteria are not in accordance with the purpose of the study. Were studies published in Arabic, English and French, carried out in any country, searched?
Page 3, lines 103-109. What is the purpose of this paragraph?
Page 4, lines 147-148. Remove percentages.
Figure 1. It has very poor quality. The figure was not made rigorously. The "eligibility" is confused with the "included".
In the text, there is no call to table 3.
Page 14, lines 209-210. The search criteria in the methods section do not show this.
Page 14, lines 221-222. The fact that most studies were published in the last ten years cannot be used as a justification for few studies in the 22 Arabic countries.
Page 14, line 232. If the studies looked at behaviours in isolation, then the research was poorly designed. What is indicated here demonstrates that the study was not carried out following its objective.
The search terms used do not allow finding articles that analyze 24-hour movement behaviours and health indicators.
Round 2
Reviewer 2 Report
Table 1. The reference must be the first column.
Author Response
The reference column has been moved to the first as requested.